# Avalon: A Benchmark for RL Generalization Using Procedurally Generated Worlds

Joshua Albrecht    Abraham J. Fetterman    Bryden Fogelman    Ellie Kitanidis

Bartosz Wróblewski    Nicole Seo    Michael Rosenthal    Maksis Knutins

Zachary Polizzi    James B. Simon    Kanjun Qiu

Generally Intelligent*

## Abstract

Despite impressive successes, deep reinforcement learning (RL) systems still fall short of human performance on generalization to new tasks and environments that differ from their training. As a benchmark tailored for studying RL generalization, we introduce Avalon, a set of tasks in which embodied agents in highly diverse procedural 3D worlds must survive by navigating terrain, hunting or gathering food, and avoiding hazards. Avalon is unique among existing RL benchmarks in that the reward function, world dynamics, and action space are the same for every task, with tasks differentiated solely by altering the environment; its 20 tasks, ranging in complexity from `eat` and `throw` to `hunt` and `navigate`, each create worlds in which the agent must perform specific skills in order to survive. This setup enables investigations of generalization within tasks, between tasks, and to compositional tasks that require combining skills learned from previous tasks. Avalon includes a highly efficient simulator, a library of baselines, and a benchmark with scoring metrics evaluated against hundreds of hours of human performance, all of which are open-source and publicly available. We find that standard RL baselines make progress on most tasks but are still far from human performance, suggesting Avalon is challenging enough to advance the quest for generalizable RL.

## 1 Introduction

A central goal of reinforcement learning (RL) is to build systems that can master a spectrum of skills in environments as noisy and diverse as the real world. While RL algorithms have matched or exceeded human performance on a number of narrowly-defined tasks such as Go and Atari [4, 7, 28, 34, 40], existing models typically fail to generalize to unseen tasks and environments, even when testing on environments drawn from the same distribution as training [11, 45, 46]. The real-world setting is far more diverse and difficult than most existing RL benchmarks; agents must seamlessly interact in a highly variable 3D environment, with no access to state information or ability to rely on hard-coded discrete actions, requiring sensory inputs and a high-dimensional action space.

---

*Correspondence to `avalon@generallyintelligent.com`.

36th Conference on Neural Information Processing Systems (NeurIPS 2022).

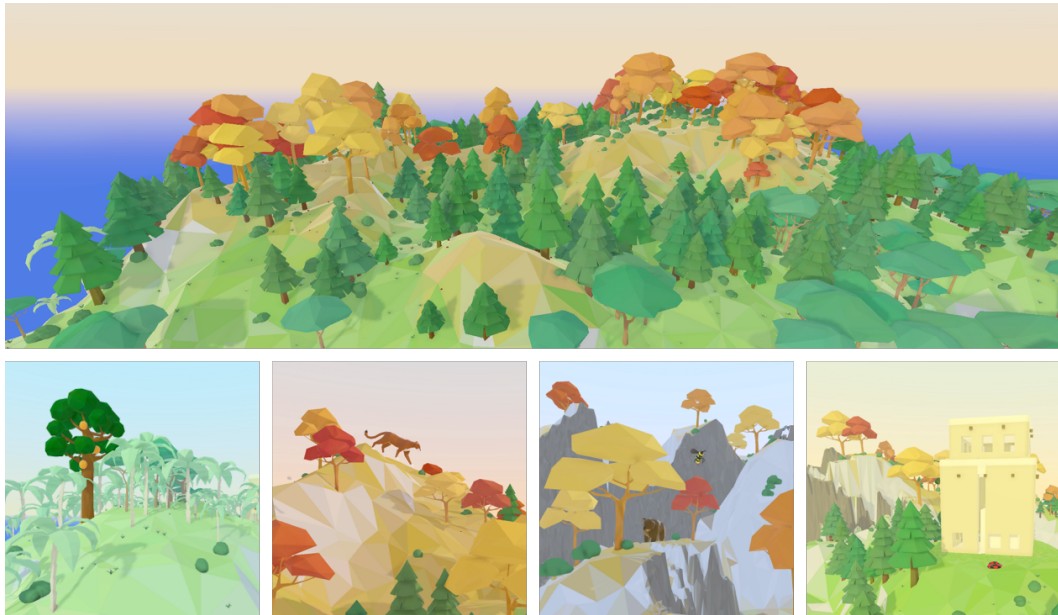

Figure 1: **Avalon is an open-world survival game requiring an agent to navigate hazards and find food.** Top: aerial view of a procedurally-generated world. Bottom row, left to right: examples of fruit to consume, predators to avoid, terrain to navigate, and buildings to explore.

To aid the study of generalization in a more realistic setting that is still tractable for current RL algorithms, we present Avalon[2]: a benchmark and high-performance simulator. Avalon is a 3D open-world survival game in which agents must navigate terrain, find food, use tools, hunt prey, and avoid predators and other dangers. Highly diverse 3D worlds are procedurally generated with many biomes, plants, items with distinct properties, and animals with unique behaviors. In any particular world, the agent's sole objective is to survive for as long as possible by finding and eating all food while avoiding hazards. Avalon agents are embodied and receive visual and proprioceptive input, forcing them to learn representations rather than relying on latent state information that would be inaccessible in the real world.

We formulate Avalon as a multi-task RL benchmark in which each task is shaped by environmental pressures rather than a task-dependent reward function. This enables all tasks to have shared dynamics and reward, making generalization more feasible compared to disjoint task spaces such as Atari's [3] where reduced or even negative transfer learning has been noted [31] [36]. Twenty tasks ranging in complexity from `eat` and `throw` to `hunt` and `survive` are included in the benchmark. A *world generator* for each task carefully alters a world so that certain skills are required to complete the task. For example, the `climb` generator only spawns food in places that cannot be accessed without climbing. Thus, Avalon's tasks are effectively a set of functions for procedurally generating environments that force the agent to learn the requisite skill to survive.

Avalon facilitates the exploration of a variety of forms of generalization:

**To unseen tasks that are structurally similar to previous tasks**. All tasks in Avalon share the same transition dynamics, action space, observation space, and simple reward based on the agent's energy level. Environment variation between tasks and within tasks is executed through the same mechanism i.e. sampling from a subset of the full distribution of possible worlds. Indeed, one could imagine widening the scope of any task-specific world generator until it encompasses some or all other tasks. This notion of task "adjacency" enables the agent to exploit shared structure between tasks.

**To unseen combinations of tasks**. In addition to 16 basic generators that map to basic skills, four compositional generators create worlds that require the agent to exercise random combinations, permutations, and variations of those skills. These compositional generators can be used to evaluate

---

[2]Named after a mythical land of Arthurian legend whose etymology is "the isle of fruit trees," since Avalon's worlds are mostly islands with fruit trees wherein the ultimate goal is to find and eat the fruit.

the agent's performance on unseen tasks that require many skills at once and provide a new setting to study the generalizability and compositionality of learned skills. In other multi-task setups, compositional tasks are often difficult to express. For example, multi-term reward functions quickly become cumbersome, and designing goal states that require complex sequential tasks is challenging.

**To unseen environments within tasks**. Unlike many popular RL benchmarks (e.g. Atari and DeepMind Control [2]) where the test environment is identical to the train environment, Avalon has significantly more variation between worlds sampled from the same distribution. Even compared to other procedurally generated benchmarks like ProcGen [8] and MineRL [15], Avalon contains dramatically more factors of variation, each of which can be individually and finely controlled. Agents in Avalon must generalize to IID sampled test environments in a setting with significantly more realism and complexity than comparable benchmarks.

Avalon's continuous action space maps to a virtual reality (VR) headset and controllers. Our choice of observation and action space allows us to measure human performance by recording demonstrations using a VR headset. We include this dataset of human VR actions, with 215 hours of human playthroughs on 1000 worlds, which may be of independent research interest.

Our benchmark includes a number of state-of-the-art RL algorithms as baselines. In Section 6, we show their performance on our tasks and highlight training mechanisms enabled by our world generation setup that help them achieve non-trivial (though still far short of human) performance.

As a final contribution, we are fully open-sourcing our environment, which represents a significant engineering effort and includes the following key contributions:

- **Efficiency**. At 7000 SPS (steps per second), Avalon is on-par with the fastest comparable simulator [38].
- **Usability**. Avalon is fully open-source, built on Godot (a free game engine with an intuitive visual editor), and includes training scripts and debugging tools.
- **Configurability**. Control over the many factors of variation in the world generators enables users to fully control the training procedure and learning curriculum.

Our hope is that the Avalon benchmark and simulator will serve as useful tools for the RL community in the quest to build more general, capable learning systems.

## 2   Related work

Avalon is the only benchmark where 1) agents learn from high-dimensional inputs in 3D procedurally generated worlds with a continuous action space, 2) the observation space, action space, transition dynamics, and reward are held constant across all tasks, 3) a very large number of factors of variation can be finely controlled in order to isolate and explore specific types of generalization, and 4) the underlying simulator is very fast and easy to use. To achieve these goals, Avalon builds on many ideas from previous works, discussed below.

Games have historically been the gold standard for benchmarking reinforcement learning methods [4, 7, 40]. The Arcade Learning Environment [3] and DeepMind Lab [2] are early examples of multi-task benchmarks that encourage mastery across several tasks. On these benchmarks, overfitting due to trajectory memorization [23] and poor or negative transfer across tasks [31, 36] remain issues.

Procedurally generated benchmarks such as ProcGen [8], MineRL [15], Malmo [19], and MetaDrive [27] aim to address per-task overfitting by varying environmental factors such as the background and placement of objects and obstacles. However, most rely on a single random seed for procedural content generation or allow users to vary only a few (usually discrete) parameters. MineRL and Malmo also run at less than 50 steps per second on a single GPU (over 100 times slower than Avalon), which makes their use for large experiments slow and expensive.

Other works such as Meta-World [44] attempt to address poor task transfer by unifying the observation space, action space, and transition dynamics across all tasks and adding controllable parametric variation of object and goal positions in order to increase shared structure across tasks. However, these environments rely on direct access to state information and include per-task handcrafted dense rewards, both of which limit applicability to the broad set of tasks that general agents are expected to solve.

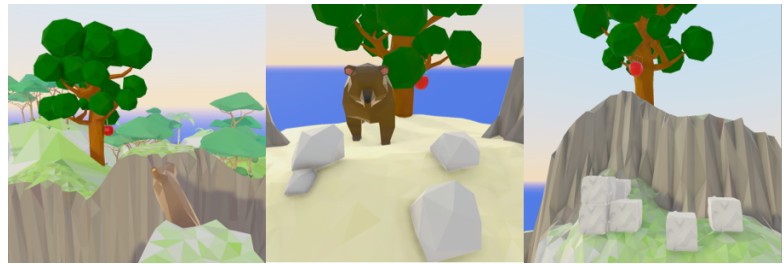 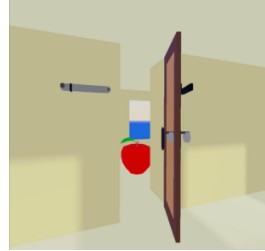

Figure 2: Example tasks `bridge`, `fight`, and `stack` (left). World generators for each task shape the environment to require completion of the task to reach food. Tasks can be outdoor or inside buildings in the environment. Buildings also enable `open` (right), where doors with complex locking mechanisms must be solved to get to food.

Other benchmarks such as Crafter [16], Obstacle Tower [22], and the Animal-AI Environment [5] avoid per-task rewards by defining a single reward function. These environments all have discrete action spaces and are limited in terms of diversity. Crafter is a 2D world; Obstacle Tower has few game mechanics: opening doors, picking up keys, pushing blocks; and Animal-AI tasks exist in a flat arena with a dozen object options.

Several other environments have substantially more diversity. MiniHack [37] is a sandbox for designing environments in the text-based game NetHack [26], which has a randomized system of dungeons with many creatures and items. However, mastery of NetHack is elusive even for humans and requires extensive game-specific knowledge, making it ill-suited to generalization research, and NetHack's symbolic inputs do not test an agent's ability to learn visual representations. By contrast, XLand [29] is a 3D multi-agent environment with visual inputs and a single reward function across tasks. Agents in XLand operate in a low-dimensional, discrete action space and are provided with explicit state information about the goals and game predicates, which severely limits the tasks that can be expressed in this constrained symbolic system. Additionally, XLand is not publicly released, so it cannot be used as a benchmark.

A separate class of related work includes photorealistic embodied AI environments such as Habitat Lab [38], GibsonEnv [43], Megaverse [32], RFUniverse [12] and BEHAVIOR [41], 3D simulators such as ThreeDWorld [13], and dataset generators such as Kubric [14]. These are valuable contributions for embodied AI and visual representation learning, but they are not designed as benchmarks for exploring RL generalization. Additionally, most of these simulators run at less than 100 steps per second (SPS) on a single GPU, which is prohibitively slow for RL research. Habitat Lab runs significantly faster (around 8,000 SPS on a single GPU), but has a limited action space, limited allowed physical interactions, and no procedural generation, making it ill-suited as a benchmark for generalization.

## 3 RL interface

The Avalon RL environment conforms to the standard OpenAI Gym environment interface [6] and is built on top of the open-source Godot game engine [21] and Bullet physics engine [9]. Below, we present some details of this environment and the design decisions that led to it.

### 3.1 Game mechanics

Avalon is a 3D open-world survival game in which players must overcome obstacles and hazards while acquiring food. Gameplay consists of a series of episodes, where each episode corresponds to a unique world, which is generated by a task-specific world generator.

A key aspect of Avalon is that special subsets of the environment distribution map to distinct tasks. Environmental pressures arising from the presence and placement of various interactable objects, enemies, and even terrain features demand the execution of a broad array of navigation and object interaction skills. For example, the existence of a deep chasm between the player and the food forces the player to create a bridge, or food may be placed in a region that is inaccessible without climbing. Figure 2 shows some examples of generated worlds for a variety of tasks.

Another critical part of Avalon is the diversity of gameplay mechanics and worlds that can be generated. There are 14 biomes, 13 plants, 20 interactable items, and 17 animals. Items include food, weapons, elements such as boxes for stacking, doors, and so on. Animals include both predators and prey, where prey are edible and can be consumed by the agent as food. All animals and food have completely distinct behaviors (for example, jaguars aggressively pursue the player and can climb trees, coconuts must experience a certain force to be opened, etc). See Appendix A for a complete description of each entity included in the game.

While the setting of Avalon is inspired by the environment in which humans evolved, it includes a number of simplifications designed to accommodate current RL systems. For example, most objects are significantly larger than their real-life counterparts because current RL algorithms are prohibitively expensive to train with very high-resolution images. As another mechanism to allow for low-resolution agents, we guarantee that food is always found near buildings or certain large trees, both of which are visible from far away.

## 3.2    Rewards

Regardless of the task, the goal of players in Avalon is to survive for as long as possible. Both human players and RL agents have a single scalar "energy" value, and when this reaches zero, the episode ends. The only way to gain energy is to eat, and thus acquiring food is the implicit goal of every level. Energy can be lost from attacks by predators or falling too far.

In order to represent these mechanics to the agent, the reward from the environment is simply calculated as the change in energy at each time step. While the reward is technically dense, in that the agent observes small changes in energy at each frame, it is effectively a sparse reward, since positive reward is attained only when the agent eats food i.e. has successfully performed the task.

To encourage efficiency and prevent erratic motion, small energy costs can be associated with movement for agents during training. More details on the exact calculations for the reward function and time limits in each episode can be found in Appendix B.

Though this (effectively) sparse reward setting is challenging, we note that dense rewards usually need to be tailored to each task and often rely on hidden state information. However, it is straightforward for a user to adapt Avalon to employ a dense reward if desired.

## 3.3    Observations

While the state data for our simulation is easily accessible, we refrain from providing any ground truth information in the observations that is not accessible to human players. The agent receives egocentric visual input in the form of a $96 \times 96 \times 4$ RGBD tensor. Just like human players, the agent is embodied and can see its own hands in its visual display. The agent is also provided with basic proprioceptive awareness: its input includes the framewise change in position and orientation for its body, the position and orientation of each of its hands (relative to its body), boolean indicators for whether each hand is within grasping range of (or currently grasping) an object, the current value of its energy, and the number of frames remaining until the episode times out. See Appendix C for the exact list of observed variables.

## 3.4    Actions

The embodied agent can move through and interact with its environment using its head, body, and hands. The primary action space is 21-dimensional and roughly maps to a virtual reality headset and controllers. It is made of $3 \times (3 + 3)$ translational plus rotational degrees of freedom, as well as $2 \times 1$ binary grasp actions and 1 discrete jump action. We also provide a reduced 9-dimensional action space which corresponds to mouse and keyboard controls. Almost all tasks can be accomplished in this reduced action space. See Appendix D for the exact action space definition.

## 3.5    Simulator

Unlike many popular RL environments, Avalon is not built on top of an existing game but rather constructed from the ground up to optimize for speed, accessibility, and, above all, scientific value. This has enabled several benefits that would otherwise be impossible:

- Avalon simulates roughly 7,000 SPS, orders of magnitude faster than most other simulators, and similar to Habitat-Sim, the fastest comparable simulator. The simulator runs without a display, making it even more efficient and easy to work with.

- Avalon was designed for the purposes of ML research, and thus benefits from a number of debugging capabilities. Firstly, it is deterministic; playing back the same set of actions from the same starting seed results in identical output. Secondly, the ground truth states of all objects can be logged continually, making it highly inspectable.

- Avalon is made using the Godot game engine [21], which is fully open-source and cross-platform (it runs on Windows, Linux, Mac, and several popular VR headsets). Godot is simple and lightweight, with a modest ∼30MB install size, and has a clean asset pipeline that supports most common formats. It boasts a full-featured visual editor, debugger and profiler that make it very easy to work with. All of our code is written in either Python or the Python-esque Godot scripting language, so it is straightforward for researchers to modify and extend, especially given the vibrant community of professional and hobbyist developers that also use Godot.

Godot is released under an MIT license, while the rest of our code is released under the GPL license.

## 4   Procedural environment generation

Avalon contains a sophisticated system for generating worlds that gives researchers fine-grained control over every aspect of variation—such as ruggedness of terrain, height of cliffs, density of predators, prey, and plants, etc.—while enabling extremely high diversity. This system empowers researchers to isolate and explore many specific aspects of generalization, ranging from the most straightforward IID setting to OOD settings like the creation of complex compositional tasks or even entirely new tasks. It also enables a simple difficulty-based curriculum that accelerates learning for our included baseline systems (described in more detail in Section 6).

Avalon supports easily scaling the diversity of the training distribution, allowing for exploration of basic IID generalization, where the test world distribution matches the training distribution. In our benchmark setting, we primarily vary terrain height and geography while leaving most diversity turned off (e.g. we don't vary colors or models of trees, landscape, food, prey, or predators, nor do we vary objects in size, shape, or weight, etc.). As RL systems become more capable, this diversity can be dramatically increased to provide more challenging IID generalization settings.

To enable investigation of generalization between similar tasks, Avalon's task-specific world generators carefully alter each procedurally generated world to require the use of particular skills. For example, the generator can raise a section of terrain to surround food with a cliff or insert an unclimbable chasm to require jumping. Our hope is that this setup enables better transfer between similar tasks, as many tasks allow for multiple solutions and the use of multiple skills, especially at lower difficulties.

In order to address a more compositional form of OOD generalization, we have designed our task-specific world generators so that they can be applied to a single world to create a sequence of tasks that must be performed. For example, our "jump" generator creates a natural-looking chasm that can only be jumped across at a certain point, while our "climb" generator creates a ring of sloped terrain that can only be climbed along a certain path. By simply composing one of these obstacle "rings" inside the other, we can create a world in which the player must accomplish both.

Our fine-grained control over diversity in world generation can also be used to directly ask about OOD generalization under small shifts in the distribution of environments. All variation in Avalon's world generation is finely controllable: the sizes and colors of all objects, the number of predators, prey, food, etc in a task, the distance of a gap that must be jumped across, etc (see Appendix E for a full list). This allows for simple, continuous ablations as any individual factor is varied from the training setting, all without any need to create or import any manual art assets.

# 5 Benchmark

## 5.1 Tasks

Avalon consists of 20 distinct tasks testing a range of navigation and object manipulation skills in diverse environments. This list includes 16 "basic" tasks (`eat`, `move`, `jump`, `climb`, `scramble`, `descend`, `throw`, `hunt`, `fight`, `avoid`, `push`, `stack`, `bridge`, `open`, `carry`, and `explore`) and four "compositional" tasks (`navigate`, `find`, `gather`, and `survive`). Each task has the same agent goal — acquiring food while avoiding hazards — but the worlds generated for each task are structured such that attaining this goal requires the skill being tested.

Worlds for the 16 basic tasks include a single food item that must be reached by overcoming at most a single obstacle. The `eat` task is the easiest: the food is created near the agent, which needs to merely grab it and bring it to its head. The other basic tasks build upon this in various ways. For example, `avoid` requires evading a predator to reach the food, `hunt` requires hunting moving prey, and `stack` requires stacking objects to reach food on a high ledge.

Worlds for the four compositional tasks are designed to require the sequential use of several basic skills: `navigate` places a series of basic obstacles between the player and the food, `find` is the same but the food is not guaranteed to be visible from the agent's starting location, `gather` has multiple fruits to find, and `survive` includes both fruit and prey animals. `survive` is a sort of "final exam" focused on breadth whose worlds can contain the elements of every other task.

See Appendix F for a complete definition of every task.

## 5.2 Training and evaluation protocols

We outline four settings in which agents can be trained and evaluated within Avalon:

- **Multi-Task, Train Basic (MT-TB)**: Train on the 16 basic tasks and evaluate on all 20 tasks. In this version of the multi-task setup, agents have seen each of the intermediary tasks but have never seen them composed together until test time.

- **Multi-Task, Train All (MT-TA)**: Train and evaluate on all 20 tasks.

- **Multi-Task, Train Compositional (MT-TC)**: Train on the four compositional tasks and evaluate on the full set of 20 tasks.

- **Single Task, Basic (ST-B)**: Train and evaluate on each of the 16 basic tasks separately.

Agents are trained with an adaptive curriculum (Section 6.1) such that the number of worlds seen in a given amount of training time is variable. While we present a particular curriculum-based training procedure, users of the benchmark are encouraged to explore alternative training procedures.

Agents are evaluated on a fixed set of 50 worlds for each relevant task, as indicated above. For example, the MT-CG and MT-S settings evaluate on all 20 tasks, and thus use the full set of 1,000 possible evaluation worlds. Tasks were randomly generated and not manually curated, except to replace levels that were reported as practically impossible. Upon investigation only one level was actually impossible, representing 0.1% of the originally generated levels. We also ensured that the evaluation worlds fully represented their underlying task generators. For example, the set of evaluation tasks for "eat" includes at least one of every possible type of food. See Appendix G for more details about the exact distribution of evaluation worlds, including the replaced worlds.

## 5.3 Human performance

To understand the difficulty of each of our tasks, we collected human player data for all evaluation levels. Approximately 215 hours of VR gameplay were recorded from 32 participants drawn from a pool of volunteers whose familiarity ranged from zero to significant experience playing similar games. For each of the $50 \times 20 = 1000$ evaluation levels, scores were averaged across at least 5 different players. Humans were provided with two practice levels for each task (not in the evaluation set), as well as basic instructions about the game mechanics (see Appendix H for details on the information provided and human data collection procedure).

Ground truth control data was recorded for each player on each level, and is available on our main website under a CC BY-SA license. We do not recommend behavior cloning from this data and testing on the fixed evaluation set; however, researchers can easily generate new evaluation levels from the same distribution for testing any such networks. The human data may also be interesting to inspect for trends, training effects, or unrelated research into VR human gameplay.

## 5.4 Performance metrics

Given a set of human player data, raw rewards can be converted into scores $S$ by normalizing such that the average human performance is $1.0$ and the performance of a random agent is $0.0$. We recommend two different aggregations of these scores. The first and more traditional aggregation is to simply take the average of these scores. The second and more robust aggregation is the optimality gap [1]. By plotting the cumulative frequency of these scores $P(S > x)$ across all runs, the optimality gap expresses how consistently the agent achieves performance above some fraction of human performance. See Appendix M for details on how scores are calculated.

# 6 Experiments

## 6.1 Difficulty curriculum

Due to the variety and scope of tasks and environments in our benchmark, we found that agents failed to learn when training on worlds sampled uniformly at random. Because the agent is effectively only given sparse positive rewards (i.e. if it finds and eats food), it is difficult for agents to make progress on more difficult tasks before improving at the `eat` and `move` tasks. For most tasks, uniform random sampling of worlds leads to a very low probability of creating worlds where random exploration policies get any reward.

As one approach to overcoming these issues, we use a simple "difficulty"-based curriculum. We employ a simple form of Automated Curriculum Learning [33] similar to the approaches in [24, 30]. When the agent succeeds (fails) at a task, the maximum difficulty of future generated worlds for that task is increased (decreased) slightly. See Appendix I for more details.

## 6.2 Baselines

We trained IMPALA [10, 25], PPO [35, 39], and DreamerV2 [17, 18] on our benchmark using the training protocols outlined above, including the difficulty curriculum. We chose IMPALA and PPO as baselines because they are among the simplest SOTA algorithms on which many other SOTA algorithms are based, and tend to be robust across problems. We chose DreamerV2 as a representative model-based baseline. DreamerV2 is used without discrete latents and without mixed precision training. No worlds are seen twice during training. Hyper-parameters were tuned via a combination of Bayesian optimization [20] and Natural Evolution Strategies (NES) [42] using runs with a smaller number of steps, and are included in Appendix J. Scores are the average of 5 runs (for 50m step results) or 3 runs (for the 500m step results). See Appendix K for details about the machines used for training.

## 6.3 Results and discussion

We show average scores for the MT-TB setting in Table 1. While all algorithms are able to achieve non-zero performance on most tasks, they fall far short of human performance, and show particularly low performance when testing on the four compositional tasks. It should be noted that non-zero performance is sometimes indicative of unexpected strategies; for example, we observe agents learning to open doors by jumping while grabbing the latch rather than moving their hand to lift the latch, thus reliably opening doors without necessarily understanding doors.

Table 1 also includes the results of longer training and ablating the curriculum learning component. The longer training shows results comparable to the 50m step training, indicating convergence of IMPALA, although learning curves in Appendix L indicate some tasks might still see improvement with longer training. Using no curriculum results in scores that are no better than random.

Table 1: Average agent scores on all Avalon tasks. Rows show scores on basic tasks (first section), compositional tasks (second section), and aggregations of tasks (third section), normalized such that mean human performance is one. The header indicates the algorithm, the total number of environment steps and whether the training curriculum is used.

| Task | PPO | Dreamer | IMPALA | | |
|------|-----|---------|--------|---|---|
| | 50m steps With curr. | 50m steps With curr. | 50m steps With curr. | 500m steps With curr. | 50m steps No curr. |
| eat | $0.708 \pm 0.067$ | $0.664 \pm 0.065$ | $0.716 \pm 0.062$ | $0.731 \pm 0.097$ | $0.001 \pm 0.001$ |
| move | $0.311 \pm 0.062$ | $0.364 \pm 0.071$ | $0.399 \pm 0.062$ | $0.409 \pm 0.076$ | $0.000 \pm 0.000$ |
| jump | $0.220 \pm 0.050$ | $0.234 \pm 0.058$ | $0.309 \pm 0.056$ | $0.287 \pm 0.072$ | $0.000 \pm 0.000$ |
| climb | $0.193 \pm 0.043$ | $0.227 \pm 0.051$ | $0.229 \pm 0.049$ | $0.332 \pm 0.074$ | $0.000 \pm 0.000$ |
| descend | $0.179 \pm 0.043$ | $0.290 \pm 0.058$ | $0.173 \pm 0.044$ | $0.222 \pm 0.059$ | $0.000 \pm 0.000$ |
| scramble | $0.306 \pm 0.054$ | $0.422 \pm 0.058$ | $0.467 \pm 0.062$ | $0.576 \pm 0.070$ | $0.000 \pm 0.000$ |
| stack | $0.091 \pm 0.036$ | $0.126 \pm 0.043$ | $0.130 \pm 0.036$ | $0.121 \pm 0.055$ | $0.000 \pm 0.000$ |
| bridge | $0.049 \pm 0.027$ | $0.121 \pm 0.045$ | $0.076 \pm 0.029$ | $0.095 \pm 0.049$ | $0.000 \pm 0.000$ |
| push | $0.113 \pm 0.039$ | $0.160 \pm 0.053$ | $0.150 \pm 0.043$ | $0.128 \pm 0.054$ | $0.000 \pm 0.000$ |
| throw | $0.000 \pm 0.000$ | $0.000 \pm 0.000$ | $0.000 \pm 0.000$ | $0.000 \pm 0.000$ | $0.000 \pm 0.000$ |
| hunt | $0.043 \pm 0.024$ | $0.063 \pm 0.028$ | $0.071 \pm 0.029$ | $0.129 \pm 0.051$ | $0.000 \pm 0.000$ |
| fight | $0.199 \pm 0.052$ | $0.336 \pm 0.076$ | $0.235 \pm 0.052$ | $0.303 \pm 0.075$ | $0.000 \pm 0.000$ |
| avoid | $0.493 \pm 0.170$ | $0.515 \pm 0.118$ | $0.603 \pm 0.159$ | $0.582 \pm 0.102$ | $0.000 \pm 0.000$ |
| explore | $0.193 \pm 0.047$ | $0.190 \pm 0.048$ | $0.213 \pm 0.048$ | $0.252 \pm 0.069$ | $0.000 \pm 0.000$ |
| open | $0.055 \pm 0.024$ | $0.126 \pm 0.041$ | $0.097 \pm 0.034$ | $0.101 \pm 0.046$ | $0.000 \pm 0.000$ |
| carry | $0.073 \pm 0.031$ | $0.066 \pm 0.028$ | $0.089 \pm 0.032$ | $0.122 \pm 0.057$ | $0.000 \pm 0.000$ |
| navigate | $0.000 \pm 0.000$ | $0.000 \pm 0.000$ | $0.012 \pm 0.010$ | $0.040 \pm 0.032$ | $0.000 \pm 0.000$ |
| find | $0.002 \pm 0.003$ | $0.000 \pm 0.000$ | $0.015 \pm 0.014$ | $0.013 \pm 0.017$ | $0.000 \pm 0.000$ |
| survive | $0.043 \pm 0.013$ | $0.044 \pm 0.014$ | $0.050 \pm 0.015$ | $0.085 \pm 0.028$ | $0.000 \pm 0.000$ |
| gather | $0.021 \pm 0.010$ | $0.021 \pm 0.012$ | $0.030 \pm 0.010$ | $0.032 \pm 0.014$ | $0.000 \pm 0.000$ |
| all basic | $0.202 \pm 0.017$ | $0.244 \pm 0.016$ | $0.247 \pm 0.016$ | $0.274 \pm 0.019$ | $0.000 \pm 0.000$ |
| all comp. | $0.017 \pm 0.004$ | $0.016 \pm 0.005$ | $0.027 \pm 0.007$ | $0.042 \pm 0.013$ | $0.000 \pm 0.000$ |
| all | $0.165 \pm 0.014$ | $0.199 \pm 0.012$ | $0.203 \pm 0.013$ | $0.228 \pm 0.015$ | $0.000 \pm 0.000$ |

Due to space constraints, the per-task optimality gap scores are reported in Appendix L (see Table 8) along with the scores from the other train-evaluate settings (MT-TA, MT-TC, ST-B) and additional results. Comparisons of the different settings (see Table 9 and Table 10) provide multiple lenses through which to view generalization. For example, we find that several tasks such as climb, descend, and avoid are too challenging for the agent to learn in the single-task setting (ST-B), yet they can be partially mastered in the multi-task settings (MT-TB, MT-TA) after pre-training on other tasks, suggesting the shared structure of the tasks provides transfer learning benefit. Furthermore, we find that agents trained on only the 16 basic tasks (MT-TB) perform as well as or better than agents trained on all 20 tasks (MT-TA), even on the compositional tasks themselves which the agent has never seen in training under MT-TB.

## 7  Limitations

While Avalon as it is today enables targeted exploration of many types of generalization in RL, there are a number of limitations. First, given the scope of the diversity possible in our environments, there are sure to be bugs and unintended results (for example, some generated worlds are impossible to complete). Another limitation is fundamental to procedural generation; while the generated worlds are quite complex, they are still far less complex than the real world, and as such, success on Avalon should not be taken as evidence that generalization is "solved" in the general case. Finally, due to limited time, we were only able to run a small number of baseline algorithms. We intend to publish updated baselines and results as they become available.

# 8 Future work

As it exists today, Avalon enables a variety of experiments, from the generalization settings highlighted here to better curriculum designs and unsupervised skill discovery. Avalon is also highly extensible by design. Users can easily create entirely new game mechanics, enabling RL research beyond the simple navigation and object manipulation tasks we have presented here.

We plan to extend Avalon in a number of ways, including by creating more tasks and more variety within each task. We also plan to include more sophisticated environment interactions to encourage tool making and tool use, as well as longer time-horizon mechanics such as rest, day-night cycles, and regrowing food to encourage the creation of RL agents that can deal with longer time horizons. Finally, we also hope to eventually include multi-agent components as well.

# 9 Conclusions

We created Avalon to address the need for a better benchmark for RL generalization and robustness. By creating a diverse set of tasks solely via environmental variation, Avalon is able to create challenging worlds that share task structure and world dynamics, hopefully encouraging the creation of more powerful and generally capable RL agents. The tasks in the Avalon benchmark are quite challenging for existing systems, despite being relatively trivial for people. Despite this, our training procedures provide a starting point with reasonable performance, on which others can improve. We hope our simulator and benchmark will serve as a foundational piece of infrastructure for future research on generalization, exploration, and other topics that are under-served by existing benchmarks.

## Acknowledgments

The authors would like to thank the following individuals for invaluable discussions and feedback on this benchmark: Anwar Bey, Tom Brown, Michael Chang, Shreyas Kapur, Andrew Lampinen, Joel Lehman, Rosanne Liu, Luke Melas-Kyriazi, John Schulman, Elias Wang, Yi-Fu Wu, and Wojciech Zaremba.

The authors are also grateful for the hard work and adventurous spirit of our human Avalon players: Meera Balakumar, Akshiv Bansal, Michael Bonanni, Andrew Cote, Rob Courtice, Dane Cross, Todd Dabney, Samkeliso Dlamini, Alejandra Encinas, Lincoln Scott Fuller, Amanda Gabbara, Nick Gabbara, Maddy Gaffaney, Gjon Gjeloshaj, Perry Goldstein, Cecilia Goss, Patrick Hoon, Schuyler Howe, Asmeret Jafarzade, Keil Miller Joseph, Luke Juusola, John Lindstedt K., Yad Konrad, Cory Li, Terrence Lucero, Kathryn Martin, Annie Melton, Brian Demeyer Michael, Kevin Multani, Jeremy Nelson, Carol Ng, Sam O'Donnell, Sam Parks, Divyesh Patel, Matthias Pauthner, Dominick Pierre-Jacques, Maria Polizzi, Nathan Ravenel, Roy Rinberg, Snigdha Roy, Katie Sapko, Phillip Seo, Brian Smiley, Derek Tam, Kaspars Vandans, Helen Wei, Christina Zhu.

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
