# OpenReview forum: "Avalon: A Benchmark for RL Generalization Using Procedurally Generated Worlds"
_NeurIPS.cc/2022/Track/Datasets_and_Benchmarks — NeurIPS 2022 Datasets and Benchmarks _

### Official Review · Reviewer_8oCL · 2022-07-20
**This paper is well written, the proposed architecture Avalon is well designed, and the benchmark could be adopted to evaluate various reinforcement learning methods.**

**Rating:** 8
**Confidence:** 4
**Clarity:** This paper is well written, and the f…

**Strengths:**

Avalon is the first reinforcement learning benchmark which allow agents to learn from a 3D simulation world. Egocentric visual input as observations instead of status vectors of instances make it more challenging for reinforcement learning method training. However, I can't tell whether visual images input will be suitable for many DRL methods.

line 62 *Agents in Avalon must generalize to IID sampled test environments in a setting with significantly more realism and complexity than comparable benchmarks.* This is an interesting
exploration. Jacob Andreas [once posted on twitter](https://twitter.com/jacobandreas/status/924356906344267776) that "Deep RL is popular because it's the only area in ML where it's socially acceptable to train on the test set." I think Avalon provides with a benchmark which evaluate RL methods with both train set and test set.

Rendering of Avalon is great. I could understand the differences between various kinds of fruits, predators and prey.

**Weaknesses:**

line 45 *All tasks in Avalon share the same trasition dynamics, action space, observation space, and simple reward based on the agent's energy level.* I think probably "Hierarchical reinforcement learning" method could avoid this challenge easily by adopting different low-level RL models with a high-level policy selector? Adopting some HRL method as baselines would be better.

**Additional Feedback:**

Are *supplemental/Appendices.pdf* and *supplemental/Avalon_Appendices.pdf* the same file? Maybe one of them is redundant? (line 567 of *Appendices.pdf* contains a single dot)

**Correctness:**

The experiment is well designed.

I think maybe a few words to explain the reasons for utilizing IMPALA and PPO as baselines will be better.

**Documentation:**

This paper is a sufficient description of the proposed benchmark Avalon. Many details are well explained in the appendices, such as observation space, action space, types of predetors and preys, details about world generation, hyperparameters of baselines training, etc.

Besides, I think the average performace of human volunteers shown in appendix might be better.

**Ethics:**

This paper provides a reinforcement learning benchmark based on simulation with no ethical concerns.

**Relation To Prior Work:**

The authors claim that Avalon is the only benchmark where
1. agents learn from high-dimensional inputs in 3D procedurally generated worlds with a continuous action space
2. the observation space, action space, transition dynamics, and reward are held constant across all tasks
3. a very large number of factors of variation can be finely controlled in order to isolate and explore specific types of generalization
4. the underlying simulator is very fast and easy to use.

**Summary And Contributions:**

This paper proposed **Avalon**, a 3D open-world survival game in which agents must perform different activities to survive as long as possible.

Avalon aims to solve the problems of generalization on previous deep reinforcement learning systems, primarily focusing on the generalization of the unseen tasks that are structurally similar to previous tasks, the unseen combinations of tasks, and the unseen environments within tasks.

---

> ### Author Response · Authors · 2022-08-11
> **Authors' Response**
>
> Thank you for your review! We appreciate your comments and are glad you liked the paper.
>
> We also thank you for your constructive and helpful feedback, which has allowed us to improve the quality of our paper. We address your key comments and questions below.
>
> ---
>
> > C1: I can't tell whether visual images input will be suitable for many DRL methods.
>
> We are interested in learning from visual inputs, which is more challenging, but state information is available and exposed to any user who prefers to learn from state vectors. **We will add a tutorial to the docs.**
>
> ---
>
> > C2a: I think probably "Hierarchical reinforcement learning" method could avoid this challenge easily by adopting different low-level RL models with a high-level policy selector? Adopting some HRL method as baselines would be better.
>
> > C2b: I think maybe a few words to explain the reasons for utilizing IMPALA and PPO as baselines will be better.
>
> We chose IMPALA and PPO as baselines because they are among the simplest SOTA algorithms on which many other SOTA algorithms are based, and tend to be the most robust across problems. **We have added a few lines to the paper explaining our choice of baselines (see bold blue text in Section 6.2).**
>
> We hope others will use the benchmark to experiment with more targeted algorithms. Your idea about hierarchical RL is very much the type of research we hope to enable with our benchmark, but implementing hierarchical RL baselines is beyond the scope of this paper. Since our baselines need to support mixed continuous and discrete action spaces as well as mixed modality inputs, implementing them is not trivial.
>
> ---
>
> > C3: Besides, I think the average performace of human volunteers shown in appendix might be better.
>
> Thank you for this suggestion. **We have added more detailed analysis of the human player data in Appendix H**.
>
> ---
>
> > Q: Are supplemental/Appendices.pdf and supplemental/Avalon_Appendices.pdf the same file? Maybe one of them is redundant? (line 567 of Appendices.pdf contains a single dot)
>
> Avalon_Appendices.pdf is the correct one, Appendices.pdf was an old copy that was accidentally submitted first - sorry for the confusion! This will be corrected in the final submission.

---

> > ### Comment · Reviewer_8oCL · 2022-08-11
> > **Response**
> >
> > Thank you for your reply. Please add those details to the paper or appendix

---

> > > ### Author Response · Authors · 2022-08-28
> > > **Update**
> > >
> > > These details have now been added (see bold blue text in Section 6.2 and Appendix H).

---

### Official Review · Reviewer_VbqH · 2022-07-21
**Good paper**

**Rating:** 8
**Confidence:** 4
**Correctness:** No concern.
**Clarity:** The paper is clearly written.

**Strengths:**

1. RL Generalization is an important topic in RL research.
2. Avalon is very challenging domain with high-dimensional input and continuous action space.
3. Baseline results with PPO and IMPALA are provided.
4. The GitHub provides docker to enable users to quickly start experiments.

**Weaknesses:**

It would be great if the authors can also provide some baselines that are specially designed for RL generalization.

**Additional Feedback:**

None.

**Documentation:**

The GitHub is well documented.

**Ethics:**

No concern.

**Relation To Prior Work:**

Previous RL environments are well discussed.

**Summary And Contributions:**

This paper presents Avalon, a new benchmark for RL generalization. It includes twenty tasks covering different skills. The implementation is efficient for RL research with 7000 SPS. The agents learn from high dimensional input with a continuous action space.  A very large number of factors can be tuned to study different types of exploration. Two RL algorithms PPO and IMPALA are provided as the baselines.

---

> ### Author Response · Authors · 2022-08-11
> **Authors' Response**
>
> Thank you for your review! We appreciate your comments and are glad you liked the paper.
>
> We also thank you for your constructive and helpful feedback, which has allowed us to improve the quality of our paper. We address your key concern below.
>
> ---
>
> > C1: It would be great if the authors can also provide some baselines that are specially designed for RL generalization.
>
> We chose IMPALA and PPO as baselines because they are the simplest SOTA algorithms on which many other SOTA algorithms are based, and tend to be the most robust across problems. We hope others will use the benchmark to experiment with more targeted algorithms. **We have added a few lines to the paper explaining our choice of baselines (see bold blue text in Section 6.2).**
>
> **New baselines:** We have also been working to implement baselines based on BYOL-Explore (Guo et al. 2022) and Dreamer V2 (Hafner et al. 2020). However, these algorithms require significant changes in order to work in our setup (supporting mixed continuous and discrete action spaces and mixed modality inputs) and we want to make sure we are implementing the algorithms to the highest standard, with zero bugs and properly tuned models. We will include them when we feel our implementations represent the algorithms fairly.

---

> > ### Comment · Reviewer_VbqH · 2022-08-13
> > **Thank you for reply**
> >
> > Thank you for the reply and look forward to see new baselines in the future.

---

> > > ### Author Response · Authors · 2022-08-28
> > > **Update on baselines**
> > >
> > > Thanks! We have just updated the paper and appendices to include an additional baseline: Dreamer V2 (Hafner et al. 2020). The results from DreamerV2 are presented in the main paper in Table 1. Additional results, hyperparameters, etc are also included in Table 7 and 8, and Figure 14. Section 6.2 of the main paper text has been updated to reflect the inclusion of this new baseline (additions highlighted in blue).

---

### Official Review · Reviewer_1zFJ · 2022-07-25
**A performant 3D environment with granular controls to enable research on generalization**

**Rating:** 8
**Confidence:** 4
**Clarity:** The paper is well written.

**Strengths:**

The most significant contribution of this submission is the Avalon simulator engine which provides diverse, customizable procedurally generated 3D tasks at a simulation speed much faster than similar competitors, making this a large step forward in accessibility.  In addition, the fine grained control over the many factors of variation in the environment and the design decisions that allow for scaling up difficulty along a variety of axes make this significant both for research pushing the performance of RL agents on challenging tasks (through e.g.  curriculum learning) and for fine-grained investigation of generalization.  The human trials contextualize the agent performance in evaluation in addition to opening up the possibility of exploration of learning from a small set of demonstrations.  The active maintenance and further development of the benchmark by the team indicates a level of commitment and attention to detail which will enable users of the environment to engage with it easily with active support, facilitating quick and correct research.

**Weaknesses:**

While this contribution is distinguished by its simulation speed and fine grained environment control, the many existing environments/datasets in this space diminish its impact on the broader research field.

**Additional Feedback:**

How difficult is it for users to add more tasks, basic or composite? -- resolved in the discussion, it is very simple to add additional tasks.

**Correctness:**

The claim of providing a performant environment are thoroughly supported through extensive profiling. The claim of providing fine grained control is supported in the thorough documentation of the optional parameters.  The claim that this environment is ideal for testing generalization is demonstrated by the experiments.  The environment design and dataset creation are sound, and the evaluation method of human performance normalized scores is appropriate.

**Documentation:**

The supporting documentation round the dataset is thorough and the plan for release and maintenance is sound.

**Ethics:**

I see no ethical concerns

**Relation To Prior Work:**

The contribution is clearly positioned amongst related work to the best of my knowledge.

**Summary And Contributions:**

Avalon addresses a gap in benchmarks for studying RL generalization by providing a 3D procedurally generated world, with a highly performant simulator, which provides control over a large number of factors of variation to allow for scaling difficulty as well as testing generalization. Additionally, it provides a set of tuned tasks which have both many similarities and differences, allowing testing of transfer and generalization. The submission provides baseline performance from IMPALA and PPO across a variety of training setups, and specifies an evaluation protocol which includes a fixed test set and human performance measures to normalize the agent scores.

---

> ### Author Response · Authors · 2022-08-11
> **Authors' Response**
>
> Thank you for your review! We appreciate your comments and are glad you liked the paper.
>
> We also thank you for your constructive and helpful feedback, which has allowed us to improve the quality of our paper. We address your key comments and questions below.
>
> ---
>
> > C1: The claim that this environment is ideal for testing generalization is demonstrated but the experiments.
>
> Firstly, we wish to clarify that there are some preliminary experiments on generalization already in the paper via the four different train-test setups (MT-TB, MT-TA, MT-TC, ST-B) introduced in Section 5.2 (with results given in Section 6 and Appendix L):
>
> - Comparing MT-TB and ST-B, we show evidence that the shared structure of tasks provides significant transfer learning benefit; for example, several tasks (e.g. jump, climb, avoid) are too challenging for the agent to learn in the single-task setting (ST-B) yet progress on them is made in the multi-task settings after pre-training on other tasks.
> - Furthermore, we show evidence of generalization to compositional tasks: agents trained on only the 16 basic tasks (MT-TB) perform just as well as agents trained on all 20 basic + compositional tasks (MT-TA) - even on the compositional tasks themselves, which the agent has never seen in training under MT-TB.
>
> **These findings are somewhat buried in Appendix L. We have expanded and moved some of this discussion into the main body of the paper for clarity (see bold blue text in Section 6.3).**
>
> Additionally, to build on our existing preliminary experiments, **we will add another train-test setup** in which agents train on each of the 20 tasks individually as in ST-B but are evaluated on all of the other (unseen) tasks. This will provide yet another condition under which we evaluate how agents generalize to new tasks.
>
> ---
>
> > C2: **Additional Feedback:** How difficult is it for users to add more tasks, basic or composite?
>
> Avalon is designed to be highly and easily extensible. Users can modify the environment through a set of Config objects that control hundreds of factors of variation (see Appendix E for detailed breakdown). This same mechanism allows users to craft new tasks, since tasks are defined and shaped by the environment generation procedure. Additionally, Avalon runs on any Godot level, meaning that users can create totally novel scenes with custom art assets or even new game mechanics and evaluate agents on them.
>
> We plan to add detailed documentation with examples showing how to modify the environment and create new tasks (including both basic and composite tasks). **This documentation will be added to the codebase before publication.**

---

> > ### Comment · Reviewer_1zFJ · 2022-08-11
> > **Response**
> >
> > Thanks for addressing my concerns, I've modified my review to reflect your answers.

---

### Official Review · Reviewer_1oJ2 · 2022-07-25
**An open-world simulator for generaliziable RL research.**

**Rating:** 8
**Confidence:** 4
**Correctness:** The experiment design is correct and …
**Clarity:** It is easy to read and interesting

**Strengths:**

I really appreciate the huge effort made by the authors to build this open-world simulator from scratch and optimize it for RL research. Current open-world simulators are mostly built on games with limited customizability and accessibility to the game's internal states, hindering the research of AGI whose goal is to build agents operating in the open world with human-like behaviors. In my opinion, Avalon will definitely speed up the development progress of this field, and the results will be beneficial to Game AI and Embodied AI.

In addition, the generalization tasks built on Avalon and research topics are reasonable and valuable, since the skill generalizability and compositionality in a sparse reward setting are important for open world agents. In the process of accomplishing the unique goal, survival, diverse behaviors and intelligence will emerge with a set of skills. Avalon is equipped with all the necessary features to support related research.

The authors provide extensive benchmark results, human behavior datasets, debug tools, and future plans, making it a qualified benchmark for publishing at the current stage.



**Weaknesses:**

1. In spite of adequate content in the appendix, the webpage is still under construction, and the document is unavailable.
2. The wall clock time means nothing for the policy learning, and GPU hours to learn a policy may vary across different GPUs, like 3090 vs A100. Therefore, I still recommend reporting the number of interactions between agent and environment instead of GPU hours. Also, sample efficiency is another critical problem in learning generalizable policies. It is a promising direction to develop algorithms to achieve higher skill compositionality with less interaction.
3. The experiments are conducted on datasets with a fixed size (50 worlds per task). Previous works like ProcGen investigate how the generalizability improves by increasing the diversity and size of training set. Therefore, more experiments should be conducted to see if increasing the number of unique worlds in each task can improve the skill generalizability.


**Additional Feedback:**

N/A

**Documentation:**

The code is released with the training script. Some basic usages and instructions are provided, but a document is unavailable now. Authors already made a commitment that a complete version will be released before the date of the main conference.

**Ethics:**

The authors follow the ethical rules strictly. The data collection is approved by IRB.

**Relation To Prior Work:**

It covers almost all related works but omits one project which also uses PG to generate environments and benchmark the generalizability of RL agents in high-dimensional state space:

Li Q, Peng Z, Feng L, et al. Metadrive: Composing diverse driving scenarios for generalizable reinforcement learning[J]. IEEE Transactions on Pattern Analysis & Machine Intelligence, 2022 (01): 1-14.



**Summary And Contributions:**

Powered by Procedural Generation (PG), the proposed Open-world simulator, Avalon, can generate diverse scenarios for studying the generalizability of RL agents. Prior works, like ProcGen, are designed for visual generalization research, while this simulator aims to benchmark the **task-level** generalizability. With the unified system dynamics, reward design, and a unique goal (survival) across all tasks/generated environments, it can comprehensively investigate the skill generalizability of embodied agents. It adopts RGB/Depth camera data with proprioceptive observation as input and enables high-dimensional continuous control, which is similar to real world setting and thus valuable for achieving embodied and generalizable AI. Unlike other open-world simulators, like Minecraft, this simulator is built on an open-sourced game engine, so that hence researchers can do more customization and access the internal states of the simulation. Also, Avalon is tailored for RL research. For example, it can run up to 7000 FPS and has a gym-style API. With these optimizations for rollout efficiency, the extensive experimental results show that it is challenging for current SOTA algorithms to reuse and combine skills for unseen tasks. This, in turn, highlights the value of the proposed simulator.

---

> ### Author Response · Authors · 2022-08-11
> **Authors' Response**
>
>
> Thank you for your review! We appreciate your comments and are glad you liked the paper.
>
> We also thank you for your constructive and helpful feedback, which has allowed us to improve the quality of our paper. We address your key comments and questions below.
>
> ---
>
> > C1: In spite of adequate content in the appendix, the webpage is still under construction, and the document is unavailable.
>
> Our documentation (here: https://github.com/Avalon-Benchmark/avalon) has been updated, and we will continue adding more documentation, including detailed examples for how to modify the environment and create new tasks. We will also continue expanding the content on the webpage (here: [https://generallyintelligent.ai/_avalon/](https://generallyintelligent.ai/_avalon/)).
>
> ---
>
> > C2: The wall clock time means nothing for the policy learning, and GPU hours to learn a policy may vary across different GPUs, like 3090 vs A100. Therefore, I still recommend reporting the number of interactions between agent and environment instead of GPU hours.
>
> We agree with your points and **we will switch to reporting training time in interaction steps rather than GPU hours**. We have also removed the line (L265-266 in the original) that recommended reporting GPU hours instead. Thanks!
>
> ---
>
> > C3: The experiments are conducted on datasets with a fixed size (50 worlds per task). Previous works like ProcGen investigate how the generalizability improves by increasing the diversity and size of training set. Therefore, more experiments should be conducted to see if increasing the number of unique worlds in each task can improve the skill generalizability.
>
> To clarify, the training sets are **not** fixed size. Only the evaluation sets are fixed to 50 worlds per task. Agents are trained with an adaptive curriculum such that the number of worlds seen in a given amount of training time is variable. This setup actually helps us train on as many worlds as possible, which the ProcGen results suggest is beneficial for generalization. Experiments to reproduce the ProcGen results can be run with this benchmark but are beyond the scope of this paper. **We have added text to clarify these points (see bold blue text in Section 5.2).**
>
> ---
>
> > C4: It covers almost all related works but omits one project which also uses PG to generate environments and benchmark the generalizability of RL agents in high-dimensional state space: Li Q, Peng Z, Feng L, et al. Metadrive: Composing diverse driving scenarios for generalizable reinforcement learning[J]. IEEE Transactions on Pattern Analysis & Machine Intelligence, 2022 (01): 1-14.
>
> Thank you. **We have added this reference to the related work section (see bold blue text)**.

---

### Official Review · Reviewer_7xx5 · 2022-07-27
**Game based benchmark based on procedurally generated games sharing same world objects, dynamics and same reward**

**Rating:** 8
**Confidence:** 4

**Strengths:**

1. The environment allows fine control of individual parameters for studying skill transfer, composability, curriculum learning and other aspects researchers might be interested in.
2. The environment is one of the fastest among the currently used frameworks.
3. Provides a set of sensory observations that are suited for studying embodied agents (POV cameras, proprioceptive sensors, orientation)
4. Procedural environment generation with ability to test skill composition.
5. Additional dataset of human gameplay of each task in the benchmark, using a VR headset.

**Weaknesses:**

1. The biggest weakness seems to be that the performance on each task is quite poor, even on tasks that are supposedly easy. It is likely due to the sparse reward, which the authors have created in an effort to have a shared reward across all tasks.
2. The benefit of keeping a common but sparse reward is unclear to me. Harder tasks of course perform worse due to the sparse reward, and hence testing current RL algorithms may all in equally poor performance, making the benchmark better suited for methods that handle sparse rewards better, rather than comparing for skill transfer or generalization.
3. The paper claims to be well suited for testing generalization. However, the dynamics are same and the tasks are largely similar, making the benchmark less diverse than others, for example Procgen.
4. The paper reports that around 2.4% of the generated tasks were unsolvable. It is not unreasonable to assume that these are likely to be the harder, composed tasks. Hence it may be the case that harder tasks are unsolvable more often.

**Additional Feedback:**

Recommend the authors to provide three things.
1. More detailed discussion as to why this benchmark is better for studying generalization as they claim.
2. Statistics of how many of each task's environments (especially the difficult and composed environments) are unsolvable.
3. Documentation and examples of how to modify the environment and create new tasks, this would help with wider adoption.

**Clarity:**

There are some missing contexts in the paper, for example. The authors state "some tasks can be accomplished in other, easier ways” without explaining what the easier way is. Also the survive task is mentioned as one of the hardest but has higher scores, probably a quirk of using human normalized optimality gap, but it somewhat calls into question the definition of a hard task. Similarly other small claims should be better explained. However, overall paper is fairly clear.

**Correctness:**

The paper uses optimality for its performance metric, which is a good metric to use in this context. The training protocol is also common and correct.

**Documentation:**

The paper mainly presents a game benchmark. Additionally it presents a dataset of human gameplay trajectories. The human gameplay data has some details provided, but no significant analysis is done on it. However this is okay since it is not the main focus of the paper.

**Ethics:**

The human gameplay dataset does not contain any personal data, hence I see no ethical concerns.

**Relation To Prior Work:**

The paper describes the differences and similarities to previous benchmarks. It also talks about its advantages and disadvantages over said benchmarks.

**Summary And Contributions:**

The authors discuss a game based benchmark they have designed based on Godot framework. It allows testing skills in an open world setting with same objects and physics across different tasks. The authors also design tasks such that the reward is shared among all tasks, but the benefit of doing so is not clear. The environment is good for testing skill transfer and composability. The easy to program interface also allows users to create their own game settings and test curriculum learning.

---

> ### Author Response · Authors · 2022-08-11
> **Authors' Response**
>
> Thank you for your review! We appreciate your comments and are glad you liked the paper.
>
> We also thank you for your constructive and helpful feedback, which has allowed us to improve the quality of our paper. We address your key comments and questions below.
>
> ---
>
> > C1a: The authors also design tasks such that the reward is shared among all tasks, but the benefit of doing so is not clear.
>
> > C1b: The benefit of keeping a common but sparse reward is unclear to me.
>
> We're interested in sparse rewards because dense rewards tend to be more hand-crafted and often rely on hidden state information (as opposed to learning directly from observables). Sharing the same reward across tasks has additional benefits for studying generalization: common structure between tasks encourages transfer learning; the task space can be easily expanded; we can specify rich *compositions* of tasks; we can evaluate and aggregate performance across many tasks without needing to compare across arbitrary reward scalings.
>
> We did experiment with a universal dense reward (distance from food) but it did not outperform sparse rewards. However, it is straightforward for a user to use Avalon with a dense reward, and **we will add a tutorial**.
>
> **We've added discussion to section 3.2 (red text).**
>
> ---
>
> > C2: The biggest weakness seems to be that the performance on each task is quite poor, even on tasks that are supposedly easy.
>
> We have performed additional tuning and run each of the baselines for 10x as long, and **will include those results**. Inspecting the agent, we actually find it has effective strategies for e.g. avoiding predators, navigating terrain, and even opening doors. **We will include some videos** demonstrating agent performance on some of the hardest tasks at which it succeeds.
>
> ---
>
> > C3a: The paper claims to be well suited for testing generalization. However, the dynamics are same and the tasks are largely similar, making the benchmark less diverse than others, for example Procgen.
>
> > C3b: More detailed discussion as to why this benchmark is better for studying generalization as they claim.
>
> ProcGen is mainly a benchmark for visual generalization, whereas Avalon (while also very visually diverse) is primarily concerned with generalization at the level of *skills*. The dynamics, actions, and rewards are deliberately unified across tasks in order study generalization within tasks, between tasks, and to compositional tasks that require reusing and combining skills learned in previous tasks.
>
> Several works have shown that in disjoint task spaces (different actions, dynamics, reward structure, etc between tasks) agents struggle to transfer learn between tasks [[a](https://arxiv.org/pdf/1511.06342.pdf)][[b](https://www.notion.so/Planning-911f86e0cfc04e748b380d505745106a)], and it's also harder to design meaningful multi-task evaluation protocols [[c](https://arxiv.org/pdf/2111.09794.pdf)]. In Avalon, world variation between tasks and within tasks is executed through the same mechanism, which imbues a sort of adjacency between tasks that enables exploitation of shared structure. This also allows us to compose tasks, providing a new setting to evaluate the generalizability of learned skills.
>
> **We've expanded this discussion in the abstract/intro (red text)**.
>
> ---
>
> > C4a: The paper reports that around 2.4% of the generated tasks were unsolvable.
>
> > C4b: Recommend the authors to provide: Statistics of how many of each task's environments (especially the difficult and composed environments) are unsolvable.
>
> These statistics were provided in the file ReplacedWorlds.txt (included in supplementary materials and referenced in Appendix G). **We will move that information into Appendix G for better visibility.**
>
> To clarify: 2.4% of the *generated* evaluation levels were unsolvable, but they were replaced with solvable levels (generated with a new random seed) such that **the agent was only evaluated on tasks that humans successfully solved.** While some of the training levels may have also been unsolvable, the adaptive curriculum makes it unlikely the agent saw those.
>
> Most unsolvable tasks were due either to bugs or to edge cases at the highest difficulty levels where the environment initialized in a way that was hard to win. We have been fixing both types of issues and are collecting more human player data. **We expect the final figure to be significantly lower than 2.4%.**
>
> ---
>
> > C5: There are some missing contexts in the paper […]
>
> Thanks for flagging, **we've added clarifications in the paper** (red text in Sec 6.3 for the first; red text in Sec 5.1 for the second).
>
> ---
>
> > C6: Recommend the authors to provide: Documentation and examples of how to modify the environment and create new tasks, this would help with wider adoption.
>
> We will add detailed documentation with examples showing how to modify the environment and create new tasks (both basic and composite tasks). **This will be added to the codebase before publication.**

---

### Official Review · Reviewer_vSwS · 2022-07-28
**An impressive open-world 3D environment for reinforcement learning targeted towards generalizability studies, with human performance data**

**Rating:** 8
**Confidence:** 3
**Clarity:** Yes, the paper is clear and well writ…

**Strengths:**

•	Good results and analysis regarding generalizability of tasks - exhibiting how the environment can be used for generalizability research.

•	 Tasks are introduced by varying terrain and objects with a random Procedurally Generated Environment. This allows all tasks to share the same dynamics.

•	Reward is not based on task – simply the delta of the energy level (which changes when the agent eats, falls, or is attacked).

•	Tuning parameters for varying the complexity of the environment.

•	Many features, e.g., 17 animals, food types, multiple key types.

•	I appreciate the attention to detail in the environment functionality. Some examples: (1) Fruit is only close to trees or buildings to aid locating, (2) animals have different activation criteria. (3) In the “throw” task smaller animals are spawned as difficulty increases. (4) Each food/animal has nuances in how it must be interacted with. (5) Multiple types of locks, some with rather intricate usage

•	I think the dataset of human results adds significant value – provides a means of normalizing RL scores against random agents (score of 0), and humans (score of 1).

•	Good supporting material, e.g., video showing the environment.

•	I feel that the code is solid. The code base is rather extensive, so it is difficult to check all facets of the work. I think it makes a valuable contribution, albeit in a very specific area (reinforcement learning for 3D open-worlds).

•	The authors continued to make significant improvements during the review cycle. It gives me the impression that the code will be well-maintained even after publication.

**Weaknesses:**

There are limited baseline results using RL agents, which the authors indicate they are working to resolve. I am satisfied with the authors' response regarding this item. I feel the authors have provided adequate justification for their baseline choices, and indicate they are working on new baselines. I agree that it would be better to make sure that they are properly implemented rather than rushing them to publication.


**Additional Feedback:**

This is simply an idea for future work – not a request: Is there an easy ways to provide a “simple” version of the environment with a simpler observation / action space. For example, by turning this into a 2D side scrolling game implementing just a subset of the tasks and features? It seems that some of the procedural generation techniques could transfer to this 2D environment, and provide a simpler environment for studying generalizability. I realize that this may require rewriting much of the code, but it’s just a brain-storming idea.

**Correctness:**



Yes, the work appears correct.
I unfortunately do not have experience with the environments in the prior work, and am not able to verify all claims. But I do not see anything obviously incorrect.


**Documentation:**

The dataset has sufficient information regarding data collection and ethics. Sufficient detail provided on benchmarks. I found the appendices very comprehensive. The authors provided artifacts such as the “Datasheet for Datasets” and human participant instructions.

**Ethics:**

I see no ethical concerns. The authors used human participants, but followed proper procedure for consent and compensation.

**Relation To Prior Work:**

Yes, the authors make a good discussion of previous work and their contribution

**Summary And Contributions:**

The authors provide an environment to serve as a benchmark for reinforcement learning. This falls under the domain open-world 3D environment. This is a challenging environment for RL algorithms: The observations are high-dimensional (provided as images of the environment), and the action space is continuous.  A number of tasks are provided. There are many features implemented: animals with varying dynamics, different types for fruits, buildings, etc..  The reward signal is based on a simple mechanic that is universal among all tasks. Although benchmark RL algorithms are limited, there is a complete dataset collected from human gameplay.

---

> ### Author Response · Authors · 2022-08-11
> **Authors' Response**
>
> Thank you for your review! We appreciate your comments and are glad you liked the paper.
>
> We also thank you for your constructive and helpful feedback, which has allowed us to improve the quality of our paper. We address your key comments and questions below.
>
> ---
>
> > C1: it would make the paper stronger if there were at least preliminary experimental results studying RL generalization using this environment.
>
> Firstly, we wish to clarify that there are some preliminary experiments on generalization already in the paper via the four different train-test setups (MT-TB, MT-TA, MT-TC, ST-B) introduced in Section 5.2 (with results given in Section 6 and Appendix L):
>
> - Comparing MT-TB and ST-B, we show evidence that the shared structure of tasks provides significant transfer learning benefit; for example, several tasks (e.g. jump, climb, avoid) are too challenging for the agent to learn in the single-task setting (ST-B) yet progress on them is made in the multi-task settings after pre-training on other tasks.
> - Furthermore, we show evidence of generalization to compositional tasks: agents trained on only the 16 basic tasks (MT-TB) perform just as well as agents trained on all 20 basic + compositional tasks (MT-TA) - even on the compositional tasks themselves, which the agent has never seen in training under MT-TB.
>
> **These findings are somewhat buried in Appendix L. We have expanded and moved some of this discussion into the main body of the paper for clarity (see bold blue text in Section 6.3).**
>
> Additionally, to build on our existing preliminary experiments, **we will add another train-test setup** in which agents train on each of the 20 tasks individually as in ST-B but are evaluated on all of the other (unseen) tasks. This will provide yet another condition under which we evaluate how agents generalize to new tasks.
>
> ---
>
> > C2: There are limited baseline results using RL agents, which the authors indicate they are working to resolve. Could you perhaps adapt agents which were developed for other 3D worlds, like MineRL or Obstacle Tower to work with Avalon? I think those could be used at least for baseline scores.
>
> We chose IMPALA and PPO as baselines because they are among the simplest SOTA algorithms on which many other SOTA algorithms are based, and tend to be the most robust across problems. We hope others will use the benchmark to experiment with more targeted algorithms. **We have added a few lines to the paper explaining our choice of baselines (see bold blue text in Section 6.2).**
>
> Regarding the baselines used in MineRL (DQN, A2C, BC, PreDQN) and Obstacle Tower (PPO and Rainbow): we used PPO, A2C is a special case of PPO ([ref](https://arxiv.org/abs/2205.09123)), the DQN methods (DQN/PreDQN, Rainbow) require discrete actions, and behavior cloning doesn't apply for us.
>
> **New baselines:** We have been working to implement baselines based on BYOL-Explore (Guo et al. 2022) and Dreamer V2 (Hafner et al. 2020). However, these algorithms require significant changes in order to work in our setup (supporting mixed continuous and discrete action spaces and mixed modality inputs) and we want to make sure we are implementing the algorithms to the highest standard, with zero bugs and properly tuned models. We will include them when we feel our implementations represent the algorithms fairly.
>
> ---
>
> > Q1: This is simply an idea for future work – not a request: Is there an easy ways to provide a “simple” version of the environment with a simpler observation / action space. For example, by turning this into a 2D side scrolling game implementing just a subset of the tasks and features? It seems that some of the procedural generation techniques could transfer to this 2D environment, and provide a simpler environment for studying generalizability. I realize that this may require rewriting much of the code, but it’s just a brain-storming idea.
>
> Thanks for the suggestion! As it stands today, there are a couple ways to simplify the environment. We have a mouse and keyboard setting corresponding to a reduced action space which is very useful for debugging. It is described in Appendix D and will be documented as part of the final release. A simpler observation space (e.g. no proprioception) is also possible presently.

---

> > ### Comment · Reviewer_vSwS · 2022-08-25
> > **Thank you for your response - I have updated my review**
> >
> > C1: Thank you for pointing this out and including some of the material in Section 6.3. I think the results/analysis in Section 6.3 and Appendix L are interesting and provide sufficient insight into this point.
> >
> > C2: This makes sense and I appreciate the intent to create high-quality algorithms.
> >
> > Q1: This makes sense, these definitely seem like good features for debugging.
> >
> > I saw a typo in Line 338: extra open parenthesis.
> >
> > Also, thank you for highlighting the changes in your paper.

---

### Author Response · Authors · 2022-06-21
**Accidental inclusion of outdated appendix file**

Please look at the appendices in the Avalon_Appendices.pdf file in the supplemental materials.

Note that an old copy of the appendices were included (as the file called "Appendices.pdf")--please disregard these.

---

### Author Response · Authors · 2022-08-20
**Additional human player data**

Dear Reviewers,

Thank you for your thoughtful feedback so far. We have made several small updates to the paper in response to your suggestions and are in the process of implementing others (see individual replies for details).

We also wish to bring to your attention that we are currently collecting some additional human player data, the results of which will be posted by EOD on August 25. If anyone is interested in the updated numbers, we encourage you to check back then.

Thank you,
The Authors

---

### Author Response · Authors · 2022-08-26
**Updated draft**

Dear Reviewers,

Please see the latest draft of our paper, which includes a full batch of new user data.

There were a few reasons that we wanted to collect additional data. First and foremost, as we continued to train improved baselines (see below) and investigate their performance, we noticed a few issues that made some of the tasks easier than were intended. We also noticed that the evaluation levels in the previous version had subtle differences between the human and agent versions. Finally, there were significant changes introduced from refactoring and bug fixing since the deadline. Given all of this, we felt it best to double check that nothing material had changed about the human performance on the benchmark by collecting another set of human data.

Nothing has materially changed about the conclusions or general trends that we saw in the original data, which gives us some confidence that the data collection procedures and benchmark evaluation is robust.

Here are some examples of the types of things that we have fixed:

- The agent was previously able to climb by grabbing and then jumping, rather than coordinating its motion
- Fall damage was sometimes not being correctly applied to the agent on descend levels
- Cherries now come in bunches and mulberries are now small (as originally intended)
- Open levels can now have multiple doors to pass through

The full list of changes will be posted in the repository. We have also:

- Removed the meta-curriculum, which did not add to the performance with the task curriculum
- Tuned and retrained PPO and IMPALA models for 50m steps each
- Ran IMPALA for 500m steps and reported learning curves (see Figure 16 in Appendix L)
- Added results for agents trained on single tasks and evaluated on (other) single tasks (see Figure 15 in Appendix L)
- Added an analysis of human performance (see Appendix H.3)
- Expanded and added discussion/clarification on various points (see individual replies to reviewers)

We have also made significant improvements in our code and documentation, but have not yet uploaded the changes to our repository.

---

> ### Comment · Reviewer_8oCL · 2022-08-27
> **Comment**
>
> Awesome 👍

---

> ### Comment · Reviewer_vSwS · 2022-08-27
> **Thanks for the update**
>
> Very nice - glad to see you are being so thorough.

---

### Author Response · Authors · 2022-08-28
**New baseline: DreamerV2**

Dear reviewers,

We have updated the paper and appendices to include an additional baseline--DreamerV2.

The results from DreamerV2 are very similar to the results from PPO, and are presented in the main paper in Table 1. Additional results, hyperparameters, etc are also included in the supplemental materials in Table 7 and 8, and Figure 14. Section 6.2 of the main paper text has been updated to reflect the inclusion of this new baseline (additions highlighted in blue)

---

### Meta-Review · Area_Chair_64Kg · 2022-09-09

**Recommendation:** Accept
**Confidence:** 4

**Metareview:**

All reviewers unanimously agreed this to be a high quality paper noting its strengths as a fast, highly paramaterisable, fully-featured proc-gen environment that is highly suited to research in a number of areas including curriculum learning, skill acquisition, and generalization.  The paper was praised for the strength of the analysis, thoroughness of supporting material and human player data.  The authors engaged with the reviewers to address criticisms in subsequent revisions and discussion, including adding more experimental baselines, and explaining why having the same reward and dynamics was not a limitation in this environment.

In the end all reviewers recommended a strong accept, and I recommend accepting this paper.

---

### Decision · Program_Chairs · 2022-09-16

Accept